# Strength Properties of 316L and 17-4 PH Stainless Steel Produced with Additive Manufacturing

**DOI:** 10.3390/ma15186278

**Published:** 2022-09-09

**Authors:** Slawomir Kedziora, Thierry Decker, Elvin Museyibov, Julian Morbach, Steven Hohmann, Adrian Huwer, Michael Wahl

**Affiliations:** 1Faculty of Science, Technology and Medicine, University of Luxembourg, Campus Kirchberg, 6 rue Coudenhove-Kalergi, L-1359 Luxembourg, Luxembourg; 2Trier University of Applied Sciences, Umwelt-Campus Birkenfeld, Campusallee, 55768 Hoppstädten-Weiersbach, Germany

**Keywords:** additive manufacturing, Charpy impact energy, fatigue properties, tensile strength, BASF Ultrafuse, Markforged

## Abstract

The number of additive manufacturing methods and materials is growing rapidly, leaving gaps in the knowledge of specific material properties. A relatively recent addition is the metal-filled filament to be printed similarly to the fused filament fabrication (FFF) technology used for plastic materials, but with additional debinding and sintering steps. While tensile, bending, and shear properties of metals manufactured this way have been studied thoroughly, their fatigue properties remain unexplored. Thus, the paper aims to determine the tensile, fatigue, and impact strengths of Markforged 17-4 PH and BASF Ultrafuse 316L stainless steel to answer whether the metal FFF can be used for structural parts safely with the current state of technology. They are compared to two 316L variants manufactured via selective laser melting (SLM) and literature results. For extrusion-based additive manufacturing methods, a significant decrease in tensile and fatigue strength is observed compared to specimens manufactured via SLM. Defects created during the extrusion and by the pathing scheme, causing a rough surface and internal voids to act as local stress risers, handle the strength decrease. The findings cast doubt on whether the metal FFF technique can be safely used for structural components; therefore, further developments are needed to reduce internal material defects.

## 1. Introduction

Metal additive manufacturing (AM) is rapidly gaining adoption throughout engineering industries, with many research resources being directed at developing new methods and usable material types. While powder bed fusion (PBF) methods such as selective laser melting (SLM) [1] and binder jetting (BJ) are well-established due to the excellent achievable part quality, they are also costly and complex in their use [2]. Advances were made to enable metal AM with more straightforward approaches centered on material extrusion methods using a filament containing polymer-metal blends similar to materials used in metal injection molding (MIM) [2,3]. MIM is a traditional process for producing high complexity parts in which powder metal mixed with binder material is shaped and solidified using injection molding. Then, the parts are subjected to a binder removal step (debinding), and finally, they are sintered to the full-density parts. On the other hand, the metal extrusion printing method is similar to the well-established fused filament fabrication (FFF) method [1,4] for polymers, also known as fused deposition modeling (FDM), with subsequent debinding and sintering steps.

In the metal FFF technology, everything starts with a CAD model that is sent to slicer software via a STereoLithography (STL) file. Next, the filament containing metal powder with polymer binder is deposited layer by layer through a nozzle onto the build plate (green part). During the slicing step, the geometry is scaled by a specific factor to account for part shrinkage during sintering. In the subsequent debinding step, the green state part is immersed in a solvent to dissolve a portion of the binder, and the debinded part is obtained (brown part). The debinding time is a function of cross-sectional area, wall thickness, and infill density; that process can take several hours to days. A grey state part is obtained after the total polymer removal by thermal debinding affected by a heating strategy [5]. In the later sintering step, the grey part is sintered to a fully dense solid part within an atmosphere of a specific gas mixture at a defined overpressure and a temperature profile. The sintering process can take several hours with the parameters depending on the sintered material.

The main benefits of the metal FFF technology are that it allows the creation of complex parts without adding powder-release channels, meaning it does not require removing unsintered, loose powder from internal cavities, and it is very cost-competitive [2]. Unfortunately, the technology also introduced constraints. Internal structures (infill) are limited by minimum infill requirements due to the material strength of brown parts. The used nozzle size determines the minimum part thickness, and the maximum thickness of any part feature is limited to prevent excessive debinding times. Aspect ratios, defined as the ratio of a feature’s maximum to minimum geometry, should be less than 8:1 [6].

Only some metal alloys are available for printing at the moment. Significant shrinkage during the sintering process from 14% to 23% was reported [7], depending on printing orientation and material. It was found that the material will exhibit anisotropic behavior [8] and have a mesh of crack-like defects related to the printing orientation [9]. Printed parts have significant porosity [3], causing low strength. In addition, a restriction on the size of printing parts limits this technology; the printer’s area, furnace volume, and sintering technology restrict the size [10].

Nevertheless, the metal extrusion AM method can help produce small parts with closed-cell infill, creating lightweight components. Therefore, the mechanical properties of common printable steel alloys (316L and 17-4 PH) can interest engineers and researchers.

The other AM method employed in the presented development is SLM technology. Selective laser melting (SLM) is a unique technology that belongs to the powder bed fusion (PBF) methods, producing objects with complex geometry from metal powders (typically in a range of 30–50 μm) [11] with mechanical properties similar to bulk materials.

SLM requires that a 3D CAD model shall be created and sent to slicer software to generate a code that controls the printer’s laser beam. The laser beam melts the powder material layer line by line, the build plate is lowered, a subsequent powder layer is deposited onto the last layer, and the powder is melted again to build the part’s geometry. These steps are repeated until almost fully dense parts are manufactured. The employed materials can be metal powder alloys, including stainless steel, tool steel, cobalt-chromium alloys, titanium, and aluminum.

The SLM process is controlled by processing parameters [12], such as laser power, scanning speed, scan line spacing, layer thickness, scanning strategy, working atmosphere, the temperature of the powder bed, and material-based input parameters.

The SLM technology is used to produce parts of complex geometric shapes, often with thin walls allowing the creation of high-strength structural elements, inaccessible for traditional mechanical manufacturing methods due to the geometric complexity. The SLM technology can be used at all stages of product development, from design concepts to low-volume production [12]. Dimension quality of the finished products is so high that the subsequent mechanical finishing processing can be neglected in some cases.

As with every manufacturing method, SLM has some disadvantages. Porosity levels can be an issue in applications where gas-tightness is crucial, for example, in high-pressure valves where leakage through the wall is unacceptable. There are constraints on the geometries of printed parts [13]; for example, the SLM method requires avoiding part overhanging with an angle limit of 45° with reference to the build platform. Often, final parts require post-processing, such as de-powdering support removal and surface grinding.

It is important to recall an essential element to understand the test strength properties of 3D printed specimens, that printed materials are anisotropic due to variation of build orientations and used printing strategies [14,15]. Therefore, predefined build orientations are typically used during printing, as shown in Figure 1, to compare different strength test data. The presented orientation nomenclature is applied in the whole text of the article.

As outlined above, the SLM and metal FFF technologies are desirable to researchers and engineers who want to create competitive next-generation products and further develop the technology. Researchers need data on the strength of the printed materials to take advantage of these brand-new printing technologies. The advantages of the metal FFF have led to intense interest in the technology in the last years, with many studies investigating the parts’ attainable mechanical properties. Extensive lists of references are presented in Tables 10–12, with the literature test data. Unfortunately, fatigue test data are rare, and the existing articles do not fully answer whether the metal FFF can be used for structural parts safely with the current state of technology.

Therefore, the authors focus on proving the hypothesis that FFF-printed steel materials have significantly worst structural properties than the same SLM-printed steel alloy, leading to the further supposition that the metal FFF technique is currently not advisable as a structural component in mechanical engineering.

In addition, the presented paper aims to fill an existing gap in the literature regarding the fatigue properties of 316L and 17-4 PH stainless steel alloys produced by the metal FFF method. So, a comparison of the mechanical properties, including strength parameters, hardness, roughness, and impact strength of materials produced in both ways, is presented to prove our hypothesis and thereby determine the scope of possible applications and contribute to the further development of FFF technology. The fatigue and impact data are of particular interest, as these mainly determine the material’s applicability in service conditions.

## 2. Materials and Methods

Two groups of specimens were built using the two aforementioned 3D printing technologies. Additionally, two SLM printers (EOS M290, Krailling, Germany, and Renishaw AM 400, Wotton-under-Edge, UK) were used to print 316L stainless steel (Table 1). Two materials were selected for the metal FFF process due to their prevalence in the metal FFF domain: BASF Ultrafuse 316L and Markforged 17-4 PH (Table 1). The BASF 316L specimens were produced on an Intamsys Funmat HT printer with parameters shown in Table 2; debinding and sintering took place in an external supplier as part of a service offered by BASF. A catalytic debinding process was used in which green parts were exposed to gaseous nitric acid (HNO_3_) in a nitrogen atmosphere and heated. The sintering process was performed in a pure hydrogen atmosphere, according to BASF.

The 17-4 PH specimens were manufactured using the Markforged Metal X system with the settings shown in Table 3. The debinding and sintering processes were performed in-house with the default parameters recommended by Markforged.

A 100% infill was applied for both materials, although a different printing strategy was employed because various slicers were used to generate the G-code. It is worth noting here that the slicer Eiger (https://www.eiger.io/) of Markforged is very restricted in terms of changing print parameters to ensure the best quality of printed parts; however, it introduces significant limitations for users.

The 316L SLM steel specimens were printed on an EOS M 290 printer and a Renishaw AM 400 printer with parameters presented in Table 4. All EOS M 290 specimens were printed by AnyShape (https://any-shape.com/), and they applied a stress relief heat treatment at 700 °C for two hours, a standard procedure for all their printed parts. The other specimens were printed onsite at the University of Luxembourg and the Trier University of Applied Sciences.

The recommended manufacturing settings for fully dense parts were used for all materials. All specimens were built vertically (ZX), the worst-case scenario for specimens in tensile loading cases, especially for the metal FFF specimens [14]. For each type of test and material, ten samples were used to balance the reliability of test results and the test costs.

Tensile testing followed the standard [19] using the tensile machine MTS 20/M with a load cell of 100 kN. A constant elongation speed of 10 mm/min and a data sampling frequency of 227 Hz were employed during all tests. The tests were conducted at an ambient temperature of 23 °C. The flat dog bone specimens have an overall length of 110 mm, 38 mm gauge length, and 3 mm thickness, as shown in Figure 2.

Axial fatigue tests were performed following the standard [20], which involves metallic materials fatigue testing axial force-controlled method with a stress ratio R = 0.1 and a constant test frequency of 30 Hz. The tests were performed using an Instron 8872 universal testing machine with a load cell of 25 kN. The tests were conducted at an ambient temperature of 23 °C. The specimens have a nominal gauge width of 6 mm, a thickness of 4 mm, a gauge length of 24 mm and a nominal length of 144 mm, as shown in Figure 3.

The surface roughness of the fatigue specimens was measured using the TESA Rugosurf 10 G roughness gauge. Measurements were made in the middle of the specimens and in the longitudinal direction on the flat and side specimen surfaces. Two parameters of the surface roughness were measured Ra and Rz.

The hardness of all fatigue specimens was measured using the HB Brinell 2.5/187.5 scale since the tested material was not surface hardened. A ball diameter of 2.5 mm was used with a proof force of 1839 N and a proof time of 15 s [21]. The hardness test was completed using an Instron Wolpert DIA-TESTOR 722 at the ambient temperature of 23 °C. The ball diameter of 2.5 mm results in a relatively large imprint of 1.3 mm on the sample’s surface, giving an average hardness result over the surface. The measurements were performed after tests of the fatigue specimens. Both fractured parts of a specimen were measured in the proximity of the fracture. Then, an average hardness value was calculated for all specimens per the tested materials. Furthermore, due to the surface texture of the printed specimens, small areas were ground manually such that the ball impression was made visible, and the impression diameters were measured using a digital microscope in two perpendicular directions.

Charpy impact tests were also performed for specimens printed in ZX orientation (vertical) with a V notch, as shown in Figure 4. The test was performed according to the standard [22] using AMSLER Pendelschlagwerk Typ RKP 450 at an ambient temperature of 23 °C. Ten specimens of each material batch were analyzed. The impact occurred on the surface lying on the opposite side of the incised notch.

## 3. Results

### 3.1. Tensile Tests

The tensile test results are summarized in Table 5 and Figure 5, and the envelope stress–strain diagrams are shown in Figure 6. The metal FFF specimens do not achieve the strength values specified by the filament manufacturers shown in Table 6. However, it should be emphasized that for the case of 17-4 PH stainless steel, Markforged published data only in (XY) build orientation, the most favorable one in the context of the strength.

For 17-4 PH stainless steel, considerable standard deviations for all strength parameters were obtained, and the large spread of the test results confirms Figure 6, where envelope curves of the obtained material characteristics are present. It should be noted that the spread of the curves is alarmingly large. The measured tensile strength of 441 MPa is much lower than the expected value of 800 MPa. The 17-4 PH batch reached elongation at a break of 0.4%, much lower than the value of 5% given by Markforged, with a significant standard deviation of 0.1%. It means that the material was very brittle and had massive properties spread. The material behavior was confirmed by studying the specimens after tests—they did not have a necking portion, a characteristic deformation before a fracture for ductile materials. The fracture of those specimens is characterized by delamination between the printed layer (see Figure 7a–e), and that failure mechanism was consistent for all 17-4 PH specimens. The fractures always happened between the printed layers (formed by printed lines), with evidence that gaps (air voids) between the layers were present after sintering (see Figure 7c–e).

Likewise, the BASF Ultrafuse 316L specimens fractured at the low elongation at break of 10.2% compared to a value of 36%, indicated by BASF (Table 6). The measured tensile strength of 314 MPa is much lower than the expected value of 521 MPa.

The yield strength and elongation at break values show considerable standard deviations, but the spread of the measured material characteristics is much smaller than for 17-4 PH specimens. The specimen fractures were typical for brittle material with a tiny necking portion. Interestingly, the fracture mechanism was the same as for 17-4 PH steel, namely, delamination of the printed layer, as shown in Figure 8. However, the phenomenon’s intensity is less visible than for the 17-4 PH specimens.

In contrast to the metal FFF specimens, the SLM samples show excellent consistency, as seen from the material characteristics’ envelope curves and low standard deviations (Figure 5). Both batches of specimens printed on EOS M 290 and AM 400 show slightly lower strength than those published by the material suppliers (see Table 6). The measured elongation at break of 38.4% for the AM 400 specimens is greater than a value of 35% given by the printer manufacturer but in the specified tolerance. In contrast, the sample produced on EOS M 290 shows the elongation at break of 38.2%, which is smaller than the published value of 54%. It should be noted that the EOS M290 specimens were heat treated.

The SLM specimens show the typical ductile steel characteristic without an evident yield strength, meaning without a plasticity range with strain hardening at constant stress (Figure 6). The heat-treated material reveals lower strength than the material as printed without increased ductility. The fracture surfaces of both batches confirm that the SLM specimens presented typical indicators of the highly ductile materials during the tensile test, namely, clearly visible necking of the specimens (see Figure 9a,b). Extensive plastic deformation (necking) was observed in all SLM specimens before fracture.

Table 5 reveals the significant difference in strength between the SLM and metal FFF samples for 316L stainless steel. Regarding the ultimate tensile strength, the discrepancy is 256.8 MPa to the detriment of the metal samples FFF. The same trend is observed for the yield strength decreasing by 290.3 MPa. Significant differences can be observed in the elongation at break; the reduction is 0.262 to the detriment of the metal FFF sample. So, the structural performance of the SLM specimens exceeds the metal FFF specimens. Even the theoretically stronger 17-4 PH steel showed lower strength when printed by the FFF metal technique compared with 316L printed by the SLM method. Additionally, the 17-4 PH samples were exceptionally brittle compared to all 316L samples.

Returning to the relationship between the curves in Figure 6, it is evident that the 17-4 PH samples show a very large scatter in the stress–strain curves. In view of the observed broken samples, it must be considered that we are dealing with the existence of partially incompletely sintered material layers. That effect can also be seen for BASF Ultrafuse specimens, but the level of the gaps between the layers is much smaller.

The best results were obtained for the SLM specimens, especially for EOS M 290 printer with the heat treatment, where the spread between 10 stress–strain curves is negligible. The rupture in the samples appears to be homogeneous in structure with very pronounced plastic deformation before fracture, as shown in Figure 9. The same type of failure was observed for all SLM specimens.

### 3.2. Roughness and Hardness Tests

Surface roughness results are presented by the two parameters Ra and Rz. Ra is the arithmetic mean deviation of surface roughness value within a sampling length, and Rz is the sum of the height of the tallest peak and the deepest valley of a profile within a sampling length. Therefore, for Rz, extremes have a much more significant influence on the final results than for Ra. Rz can be used to check whether the profile has protruding peaks that might affect a part function, and Ra is meaningful for stochastic surface roughness of machined parts.

Table 7 and Figure 10 illustrate the results of a measurement of the surface roughness of the fatigue specimens. Two measurements were made on a flat and a side along the sample. The measurements were in the center of the specimens. As shown in the table, the results indicate that the metal FFF samples exhibit higher roughness values than their SLM counterparts in both measurement locations. For the flat orientation, the SLM 316L samples had approximately a mean Ra roughness of 4.1 µm, the BASF Ultrafuse 316L samples reached a mean Ra of 7.3 µm, and the Markforged 17-4 PH batch had a mean Ra of 8.2 µm. For the side orientation, the SLM 316L samples had a mean Ra of 4.8 µm, while the BASF Ultrafuse batch had a mean Ra of 7.5 µm, and the Markforged 17-4 PH specimens were as high as a mean Ra of 16.6 µm. The high Rz values of the FFF metal samples indicate that high surface profile extremes characterized their surfaces.

Table 8 contains the harnesses measurement of the fatigue specimens. The highest hardness was measured for 17-4 PH samples with a mean of 261 HB and the lowest for BASF Ultrafuse 316L samples with a mean of 126 HB. The most significant standard deviation of 17.4 HB for the ten samples tested was for 17-4 PH. The measured hardness values mean that all materials are soft.

### 3.3. Fatigue Tests

The results of the fatigue tests are presented in Figure 11 as an S-N plot on a log-log scale. The test results for all group specimens are at least highly correlated; in other words, the data with their group closely resemble a power trendline. What is striking in this figure is the curve for EOS M 290 samples. One test point (22,762,694 cycles; 341 MPa) deviates significantly from the power trendline trend of the other points; this is due to an insufficient number of test samples in the 1–2 million cycles range.

The graph shows that the greatest fatigue strength is for the SLM 316L specimens printed on the EOS M 290 printer and then heat-treated, whereas the worst is for the BASF Ultrafuse 316L specimens printed via the metal FFF method. The difference is significant; for example, for 1 × 10^6^ cycles, the difference is 87 MPa. What stands out in this figure is the different curve slopes for those specimens. Interestingly, the slope of the SLM 316L specimens printed on the Renishaw AM 400 printer has a curve slope one order of magnitude larger.

Further analysis of the roughness in Table 7 and Figure 10 reveals that the EOS M 290 316L specimens had the lowest average Ra roughness value of about 4 µm, while for BASF Ultrafuse 316L, Ra was 7.5 µm. Furthermore, the hardness between those specimens was also different, with 213 HB for the EOS M 290 batch and 126 HB for the BASF Ultrafuse specimens see Table 8. Comparing the two results confirms the well-known tendency for fatigue strength of materials to increase with decreasing roughness and increasing hardness. The results also indicate that the lower fatigue strength for the Renishaw SLM specimens is influenced by the mean Ra roughness value of 5.8 µm and lack of heat treatment after printing which is usually made to remove internal residual stresses (stress relief) [24].

Looking at Figure 11, it is apparent that the 17-4 PH specimens have a low fatigue strength compared with the tested SLM 316L specimens, which is a surprise considering the tensile strength shown in Table 6. The most striking result to emerge from the data in Table 7 is that very high Ra roughness of 16.6 µm measured on a side along with the sample for the Markforged 17-4 PH specimens, which is 2.21 times greater than for the BASF Ultrafuse ones. Such difference in the roughness directly influences the fatigue strength results.

Further analysis of a fracture of the Markforged 17-4 PH specimens shows that the fracture mechanism is based on delamination of the printed layers, as seen in Figure 12b,c. For the metal FFF specimens, beach (clamshell) marks were not observed on fatigue fracture surfaces. Macroscopically, the fracture surface is flat with evidence of the printing layers’ delamination and perpendicular to the applied stress. The presented specimen does not show necking; therefore, the fracture may be considered a brittle fatigue fracture. It is incontestable that there were voids between the printer layers, which were sources of crack initiations. All specimens from this group had the exact failure mechanism. An analogous fracture mechanism was detected in the Ultrafuse 316L specimens but with less evident delamination between the printed layers (see Figure 13). Those specimens had an almost imperceptible necking part. Furthermore, it should be noted that the fatigue fracture surfaces for all metal FFF specimens are like the tensile test failure surfaces.

Turning to the experimental evidence on the SLM fatigue specimens (Figure 14 and Figure 15), the fatigue fractures are not flat looking at them macroscopically. They have a flat portion perpendicular to the applied stress and the other part with a very expressive necking. The flat parts originated during fatigue cycles–fatigue zone *A_f_*, and the other part *A_c_* is the remaining material fractured catastrophically (final fracture). The ratio between those areas depends on the applied stress level. If the ratio, $A_{f}/A_{c}<1$ it is a case of the fracture in the low cycle fatigue regime, whereas when the ratio, $A_{f}/A_{c}\geq 1$ is the fracture in a high cycle fatigue regime [25]. As seen in the Renishaw AM400 specimen, the ratio $A_{f}/A_{c}=3.62$ corresponds to 1,492,546 cycles, whereas for the EOS M 290 specimen, the ratio is 1.13, corresponding to 392,641 cycles.

Additionally, beach (clamshell) marks were not observed on fatigue fracture surfaces. One must remember that the non-occurrence of beach markings means continuous crack growth during load cycling, which is common in fatigue tests of samples at constant load amplitudes. Therefore, while it is reasonable in many cases to identify fatigue as a cause of failure based on beach markings on the fracture face, this should not always be completed [25]. Due to the lack of beach marks, it is challenging to recognize the location of the crack origin; however, on closer analysis of the fatigue fracture face, it is possible to approximate the crack origin based on the radial groove pattern.

For both groups of the SLM specimens, the described failure mechanism was observed, as one can see in Figure 14 and Figure 15. One difference between the two is the noticeably larger necking part for the EOS M 290 samples than for the Renishaw AM400 316L samples, apart from the apparent discrepancy between the test stress levels in the presented specimens that manifests itself in different values of the ratio $A_{f}/A_{c}$.

### 3.4. Charpy Test

The metal FFF specimens of both steel types failed to generate an unmeasurable impact resistance; hence, no results can be reported. The fractures were confined to the printing plane, indicating poor inter-layer bonding. Some portions of the adjacent layer (printing lines) experienced sufficient cohesive strength to be not detached from their original layer. No apparent plastic deformation of the cross-section is visible. One can see a distinct printing pattern on fracture surfaces (Figure 16).

In contrast, the SLM specimens showed significant resistance and ductility, indicating good layer bonding. The absorbed impact energy was 202 J (standard deviation: 16.4 J) for the Renishaw specimens and 223 J (standard deviation: 17.3 J) for the EOS specimens. The samples demonstrated fractures with large plastic deformations and rough surfaces (see Figure 17).

## 4. Discussion

### 4.1. Test Results

The presented test data support the hypothesis that the metal FFF printed steel materials (316L alloy) have a significantly lower structural performance than the same steel alloy printed using the SLM method. A yield strength reduction of 59% compared to the heat-treated material was determined, and a 65% reduction in comparison with the non-heat-treated material. A drop in the tensile strength of 41% is observed compared to the heat-treated material and 45% to the non-heat-treated one. The same trend can be seen for the elongation at break. The reduction is 73% compared to heat-treated and non-heat-treated materials, meaning that the material printed via the metal FFF loses ductility significantly due to insufficient bonding of the printed layers observed after a fracture.

Another important discovery is that the fatigue strength for the FFF and SLM specimens of 316L stainless steel differs significantly. For example, for 1 × 10^6^ cycles, the discrepancy is 56% between the metal FFF and heat-treated specimens and 34% compared to the non-heat-treated one, to the detriment of the metal FFF process. What is surprising is that the proposed heat treatment drastically improves the fatigue strength of the SLM specimens. Looking at surface roughness in Table 7, one can see that the metal FFF 316L specimens had a maximum Ra of 7.5 µm and the SLM specimens had a maximum Ra of 5.8 µm. This greater roughness led to a lower fatigue life as well.

The current study found a significant difference in the repeatability of test results to the detriment of the metal FFF technology. The low repeatability for the metal FFF specimens can be seen particularly evident in Figure 6 with the maximum and minimum enveloped curves of the material characteristics and in Table 5, looking at the standard deviations. This finding is valid for all metal FFF specimens.

The most unexpected finding is that the internal defects (voids) between the printed material layers formed by the printed lines determined the strength of the metal FFF samples so severely. The authors expected that effect but not to the measured extent. The phenomenon (delamination of the printed layers caused by air gaps between the extruded lines [7,14,26] while loading) is striking in all test types carried out and for all specimens. This was most evident in the Charpy tests, where no results could be obtained due to the specimens’ negligible impact resistance (Table 9). For the SLM specimens, this problem did not exist. Consequently, the greatest energy absorption was measured for the Renishaw AM 400 samples at 223.3 J, which is 9.4% more than for the EOS M 290 specimens, and both sample groups showed typical fracture for ductile materials with large plastic deformation, as revealed in Figure 17.

The degrading effect of high roughness on fatigue strength properties is well known, so roughness control is crucial for structural components. This is also reflected in the standard [20], listing the roughness as a factor influencing fatigue test results. For this reason, our fatigue test samples were also subjected to the roughness test.

The current study found that the surface Ra roughness for the metal FFF 316L specimens is much greater than for the SLM specimens, with a maximum measured difference of 101%. Moreover, the surface roughness can vary much between the metal FFF printers. For example, for the Markforged specimens, the roughness measured on the side of the specimens and along a sample was Ra = 16.6 µm, and for the BASF Ultrafuse samples, only 7.5 µm. A possible explanation for this might be those different printing parameters, printing strategies, and filaments. An example of the surface texture of the specimen can be seen in Figure 7c. A high value of Rz for those FFF specimens confirms the presence of high extremes (peaks and valleys) on their surfaces.

The most surprising aspect of the roughness data is a significant difference in the roughness for the Markforged 17-4 PH specimens depending on the measurement locations. The measurement performed on a flat, along a sample, gives a mean Ra of 8.2 µm but on the side, along a sample, gives a mean Ra of 16.6 µm. An explanation for this might be the type of filament properties used and the printing strategy. The build orientation may also come into play, as thin and tall structures tend to wobble during printing as they are dragged sideways by the lateral forces exerted by the nozzle, resulting in an uneven surface structure. The drag forces may vary drastically depending on the layer height, material properties, and printing speed.

The very high roughness of the metal FFF samples means that the fracture initiation occurs very quickly, which translates into reduced fatigue strength. This trend is seen in our test results.

The tests made for 17-4 PH stainless steel printed using the metal FFF confirms all findings for the metal FFF 316L samples. The material is very brittle with low strength caused by the presence of internal defects (Figure 7)—voids between the printed layers, which is more evident than for 316L. Those defects also cause low fatigue strength (Figure 12). Due to the randomness of the printed layer delamination, the material’s properties have varied greatly, as shown in Figure 6—the large scatter taken up by the minimum and maximum envelope curves. Thus, the most prominent finding to emerge from the test results is that applying the metal FFF technology for structural parts is risky now, but the technology still offers a high potential for further development in the direction of reducing the internal defects of the printed parts.

Contrary to expectations, this study shows a significant difference between the E- modulus for the SLM specimens printed using the different printers (see Table 5). The measured E-modulus for the EOS M290 specimens was 160 GPa, which was 19% greater than for the AM 400 specimens. The heat treatment of the first samples could explain a slight difference, but the 19% variation is unusual. One would have to focus on determining the modulus precisely and see what causes such a significant difference in results. The authors presume that a discrepancy in porosity can explain the observed effect. Another difficulty is the limited literature data on the E-modulus with which a comparison can be made (Table 10).

### 4.2. Comparison with Literature

#### 4.2.1. Tensile Test Considerations

Comparing results from metal FFF and MIM specimens is particularly interesting, as the base material composition and the debinding methods are similar [4]. The tables (Table 10 and Table 11) show that the printed materials are anisotropic and have lower strength and E-modulus than MIM materials. The elongations at break measured in ZX builds orientation show that the printed material is much more brittle than MIM. This phenomenon is explained in the metal FFF technology by the presence of structural defects perpendicular to the layer direction (Z) [4]. The authors [7] deduced that the bead’s orientation perpendicular to the layer direction (Z) increased the average metal particle distance in the layer direction, which could also cause the higher linear shrinkage in this direction. Additionally, if the metal particle distances between those layers are too large, it would cause voids after the sintering process. During tensile tests, these voids oriented perpendicularly to the tensile direction act as stress concentrations leading to poor mechanical properties. For the SLM process, it can be explained that weak interfacial layers for vertically built samples are parallel to cracks, providing more accessible paths for shear bands coalescence and void growth under tension loading than horizontally built ones [36]. Additionally, it has been reported that specimens fabricated in the vertical direction typically contain extensive porosity compared to horizontal directions [37]. Therefore, the presented investigation focuses on analyzing the built-in ZX direction specimens, making a conservative assessment of the mechanical properties of the printed materials.

As it can be seen from the presented literature data, the porosities of the specimens produced by metal FFF are greater than those created by SLM, which can be one of the elements causing the much lower strength of the metal FFF specimens. An unfavorable shape of structural defects and their distribution in the samples reduce the mechanical properties of the FFF samples. As reported [26], the existence of pores showed a significant impact on tensile fatigue strength because large pore-induced voids that contain subcracks near the surface of the sample contribute to the fast failure of the tensile fatigue specimen.

The reported literature data shows that for 316L steel, the best tensile strength results are achievable for SLM in XY build orientation with a slightly lower value of 561 MPa, than for wrought material with 621 MPa, although the largest elongation at the break is reported for the wrought material at 59%. The tendency for the 17-4 PH stainless steel is different in the relevant literature. The highest tensile strength of 1068 MPa (without heat treatment) is stated for metal FFF in the XY build orientation, similar to SLM and wrought material, while the greatest elongation at the break is reported for the SLM sample at 61%. However, the elongation at break for all metal FFF samples is much lower than for other specimens. The lowest strength values with minimum elongation at break are reported for high porosity samples for both materials. The literature data for both materials reveal that MIM specimens show closer strength results to metal FFF specimens, although MIM specimens are more ductile and have high E-modulus comparable with wrought material. The tensile test results of 316L specimens manufactured with SLM show higher yield and tensile strengths than their metal FFF counterparts, while for the 17-4 PH specimens, the yield strength is comparable with the SLM and MIM samples and much lower than for the wrought material.

The variation in the physical properties of the specimens produced by different technology sections can be explained in terms of microstructures developed in the considered manufacturing processes [35]. The microstructure is a very broad concept and includes porosity, pore shape, crack density, dislocation density, grain size, etc. Due to a massive variety of microstructures between those specimens, particularly in porosity and type of internal structural defects, a significant difference in strength is observed.

The test results (Figure 6 and Table 5) are in accord with recent studies (Table 10 and Table 11), indicating that the materials 316L and 17-4 PH printed using the metal FFF process have much lower strength in the context of yield and tensile strength than the material printed using the SLM method. The elongation at break results also supports previous research, which shows a significant reduction of ductility of the material printed by the metal FFF method due to internal defects and porosity. In contrast, the elongation at the break of 0.4% revealed in the 17-4 PH stainless steel study is lower than those found in the literature, but within the range of values presented by other authors. The obtained test results of the E-modulus for the metal FFF specimens do not differ much from the literature data.

Consistent with the literature, the material printed using the SLM method shows excellent mechanical properties, which can be compared with wrought material (Table 5, Table 10 and Table 11). The obtained strength properties are in a range reported by other authors.

#### 4.2.2. Fatigue Test Considerations

An overview of the fatigue properties of 316L and 17-4 PH alloys is provided in Table 12. The fatigue behavior of the materials as wrought and from the metal FFF and SLM manufacturing methods has been evaluated thoroughly, but studies investigating fatigue of MIM 17-4 PH appear to be rare in the literature. Moreover, differing manufacturing, post-processing, and testing conditions render direct comparisons difficult.

No study has investigated the fatigue properties of 17-4 PH stainless steel produced using the metal FFF method. Only one paper concerning fatigue testing of BASF Ultrafuse 316L (metal FFF) at three tensile stress levels is available at the time of writing, in which the endurance limit is determined to lie between 80–100 MPa at R = 0.1 [26]. The authors indicate the build direction XY of the specimens with a porosity value of 4.4%. They explained these low tensile fatigue strength results as partially due to the unmachined rough surface that facilitates the fast crack creation as well as the existence of internal pores. Indeed, as-built metal FFF specimens exhibit a higher surface roughness as built than SLM and MIM specimens [2,31,35].

Similarly, the porosity is typically the greatest for metal FFF than SLM and MIM, as seen in Table 12. Porosity and roughness data induce that the best results for the endurance limit are achieved for wrought material for both analyzed alloys. As can also be seen in the presented overview, the heat treatment significantly increases the endurance limit for SLM and wrought specimens.

In summary, comparing all the mentioned studies, it is apparent that these significant variations of the mechanical properties are a function of the printing parameters, which confirms the finding from [53]. It can be concluded that the fatigue performance of 3D printed parts depends on the quality of the microstructural morphology, which is driven by porosity, distribution of pores and defects, and their shapes. The resulting material imperfections depend on the printing parameters and the used material. Therefore, wrought material performed the best in reviewed articles (minimum defects), with the machined/HIPed SLM specimens being close. Hot Isostatic Pressing (HIP) post-treatment is an exciting method to reduce porosity. It improves the mechanical properties and microstructure [53] of the 3D printed parts by applying high isostatic gas pressure at elevated temperatures.

The existing limited literature data do not allow a direct comparison of the test results obtained for metal FFF and SLM samples. The problem is a lack of literature data or different test and material conditions. However, as can be seen, the results obtained for SLM samples (Figure 11) do not differ significantly from those observed by other authors (Table 12). The test results also show the tendency to increase fatigue strength with heat treatment or surface roughness reduction, similar to the literature data.

### 4.3. Final Assessment

The presented study focused only on two materials, so that the comparisons may be somewhat limited. However, the whole spectrum of tests was conducted for ten specimens per test and per material. This approach gives confidence to the obtained results. To produce specimens, the authors used two different printers per the employed technologies to see an influence of a particular printer. To enhance the outcome of the analysis, it would be beneficial to add the measurement of the porosity by employing CT (computed tomography) scanning of the specimens so the internal defects can be seen and assessed. Unfortunately, this was not possible for the presented project.

The presented results and observations support our hypothesis that FFF-printed steel materials have significantly worse structural properties than the same SLM-printed steel alloy. The main reason for this is the variability of the material properties and the meager impact resistance caused by the characteristics of the metal FFF technology. The findings cast some doubt on whether the metal FFF technique can currently be safely used for structural components. In this case, a better solution is to use SLM technology instead.

## 5. Conclusions

This study aimed to evaluate the properties of the material (BASF Ultrafuse 316L stainless steel) printed using the FFF method and to compare it with the material produced through SLM. Additionally, 17-4 PH stainless steel specimens made via the metal FFF additive manufacturing (Markforged Metal X) were tested to determine the mechanical properties.

The findings clearly indicate that due to the low repeatability of material properties, and low impact resistance, we currently do not recommend using the metal FFF method to produce any structural parts.

Material printed using the SLM method shows excellent mechanical properties, which can be compared with wrought material. We recommend applying a heat treatment for the SLM parts after printing to improve fatigue resistance. Moreover, the static strength parameters of 316L stainless steel printed via the SLM technique are surprisingly repeatable.

The findings indicate that further development of the metal FFF technology is required to improve the connection between the extruded lines by eliminating air gaps during printing and optimizing the sintering process. Those gaps are a cause of inferior mechanical properties, especially evident in the printed orientation ZX. In the case of 316L stainless steel, a yield strength reduction of 59% compared with the SLM heat-treated specimens and 65% with the non-heat-treated specimens was determined. A tensile strength drop of 41% was measured with the heat-treated samples and 45% with the non-heat-treated samples.

Fatigue strength of 316L is lower for the metal FFF specimens than for the SLM ones; for 1 × 10^6^ cycles, the discrepancy is 34% between the metal FFF and SLM non-heat-treated specimens. For the SLM heat-treated specimens, the difference is more significant and reaches the value of 56%.

Additionally, the scattering of tensile test results of the metal FFF specimens is substantial due to the randomness of the occurrence of internal defects. Furthermore, those defects cause the metal FFF specimens to have no measurable Charpy impact resistance when printed in ZX orientation.

The strength results are abysmal for 17-4 PH stainless steel, where internal defects make the material weak (low static and fatigue strength) and very brittle, with significantly varying material properties caused by internal defects.

The mean Ra surface roughness of 7.4 µm of the 316L FFF specimens was more significant than the SLM specimens, with a mean Ra of 4.4 µm. The Markforged 17-4 PH specimens had a maximum mean Ra roughness of 16.6 µm. Moreover, a significant Ra roughness difference was observed depending on measurement locations ranging from 8.2 µm to 16.6 µm, most probably caused by printing parameters and the build orientation.

## Figures and Tables

**Figure 1 materials-15-06278-f001:**
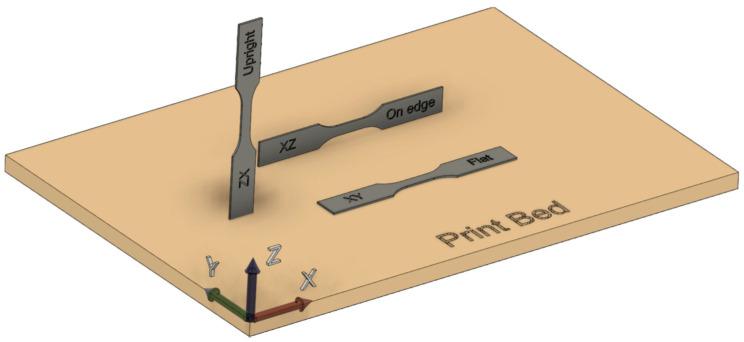
Printed specimen and predefined build orientations.

**Figure 2 materials-15-06278-f002:**
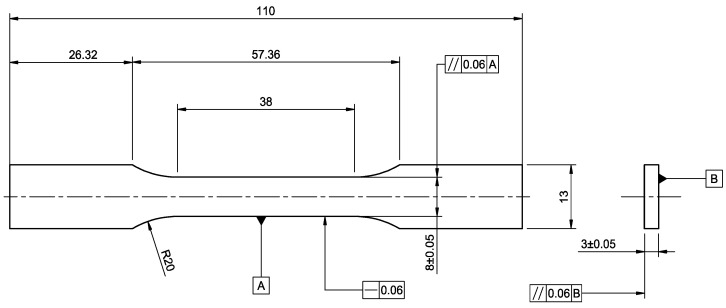
Tensile specimen design according to the standard [19], units: mm.

**Figure 3 materials-15-06278-f003:**
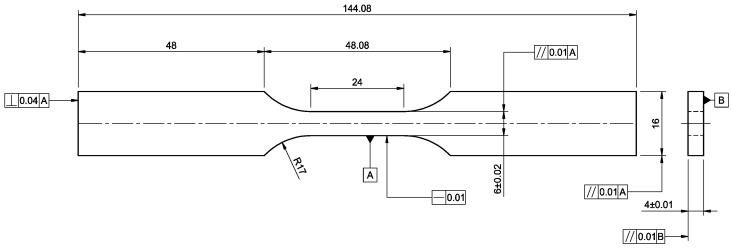
Fatigue specimen design according to the standard [20], units: mm.

**Figure 4 materials-15-06278-f004:**
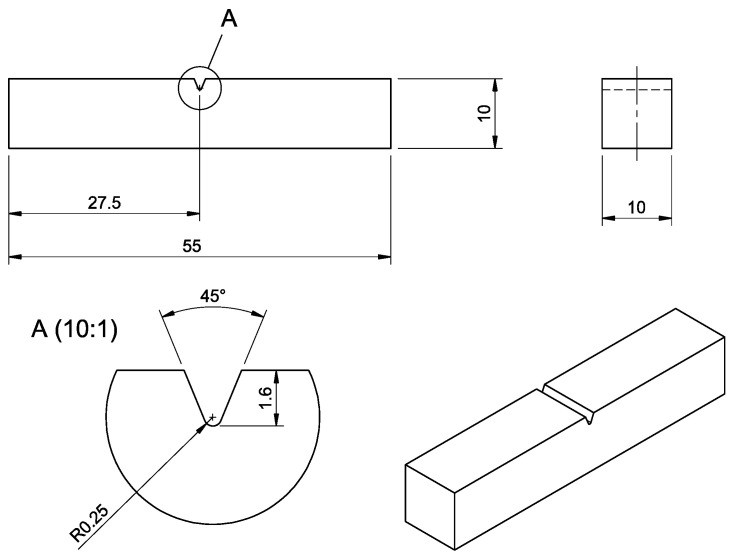
Charpy specimen design with the V notch according to the standard [22], units: mm.

**Figure 5 materials-15-06278-f005:**
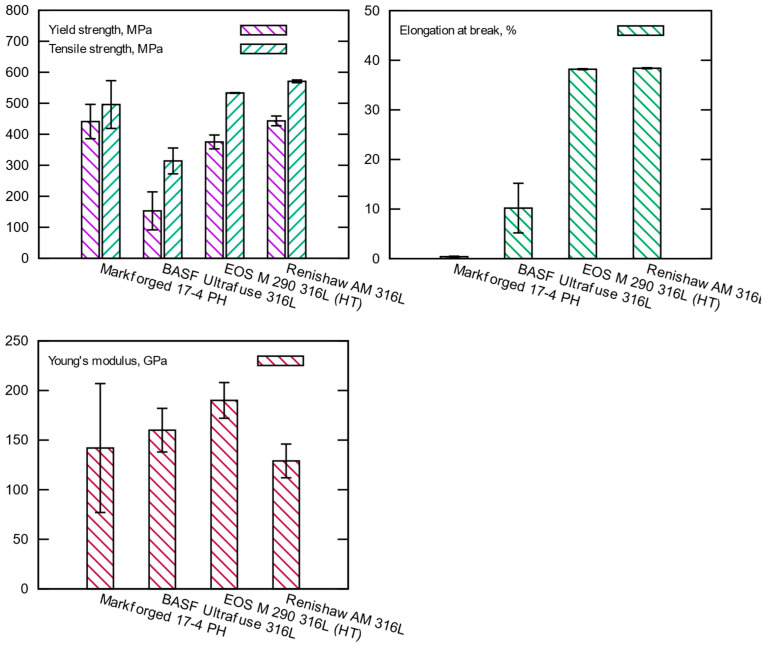
Results of tensile tests (mean with +/− standard deviation).

**Figure 6 materials-15-06278-f006:**
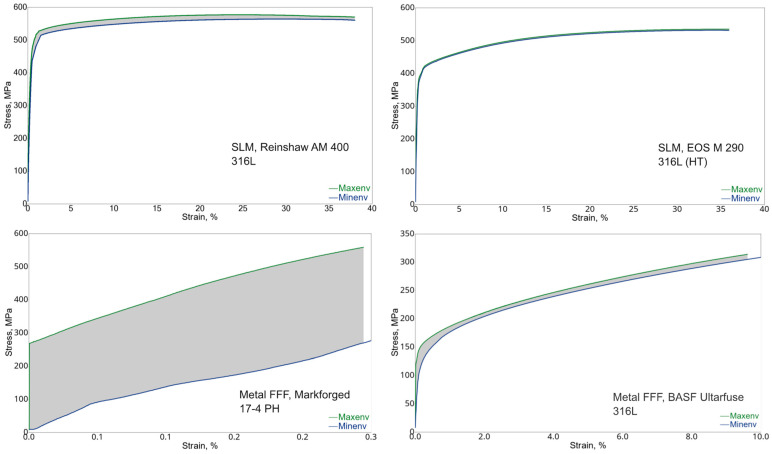
Envelope stress–strain curves (engineering stress versus engineering strain). Plots were created based on performed tensile tests considering all specimens.

**Figure 7 materials-15-06278-f007:**
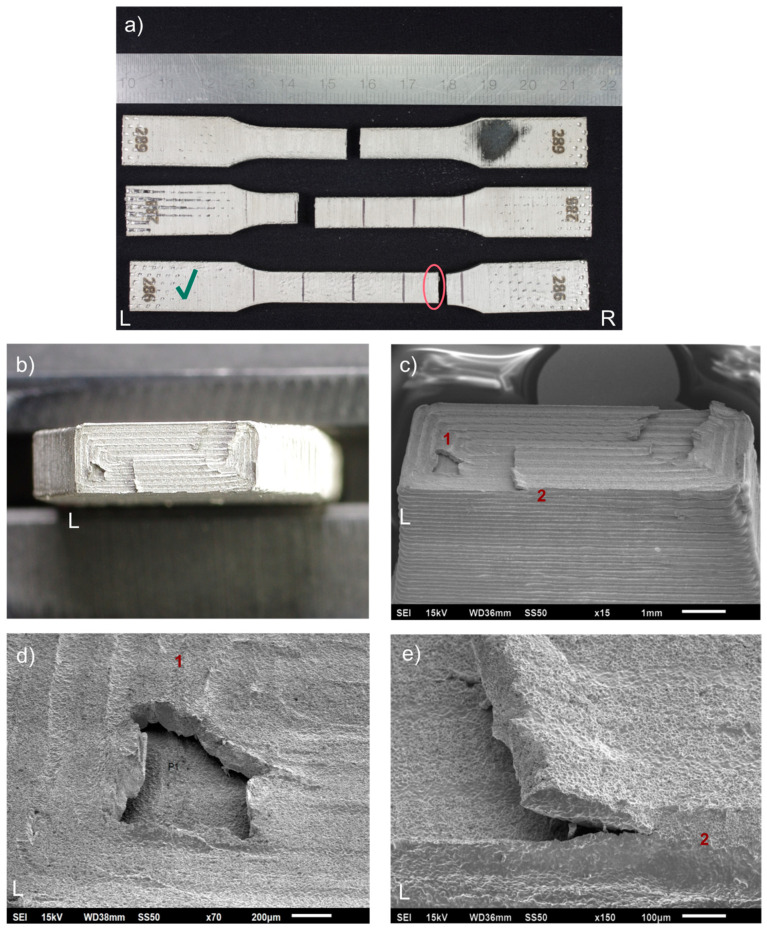
Fractures of tensile test 17-4 PH specimens presented as: (**a**) fractured specimens; (**b**) fracture surface, left side (optical microscope image), specimen 286; (**c**) fracture surface, left side (SEM image), specimen 286; (**d**) local delamination of a printed layer in the fracture, feature 1 (SEM image), specimen 286; (**e**) local delamination of the printed layer, feature 2 (SEM image), specimen 286. The samples have been numbered, which are visible in the figure. The letters L and R stand for the left-hand and right-hand sides of the sample in the figure. A green check mark means the specimen whose fracture faces were shown.

**Figure 8 materials-15-06278-f008:**
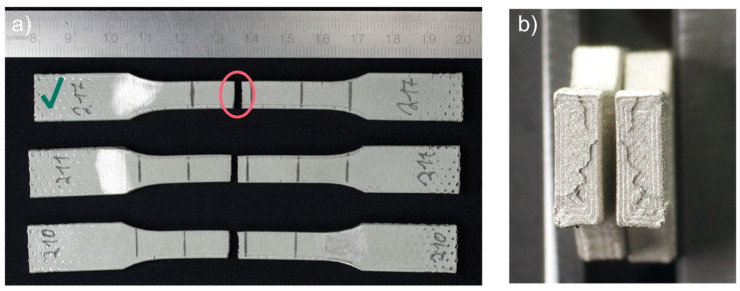
Fractures of tensile test BASF Ultrafuse 316L specimens presented as: (**a**) fractured specimens; (**b**) optical microscope image of the fracture surfaces, specimen 217. The samples have been numbered, which are visible in the figure. The letters L and R stand for the left-hand and right-hand sides of the sample in the figure. A green check mark means the specimen whose fracture faces were shown.

**Figure 9 materials-15-06278-f009:**
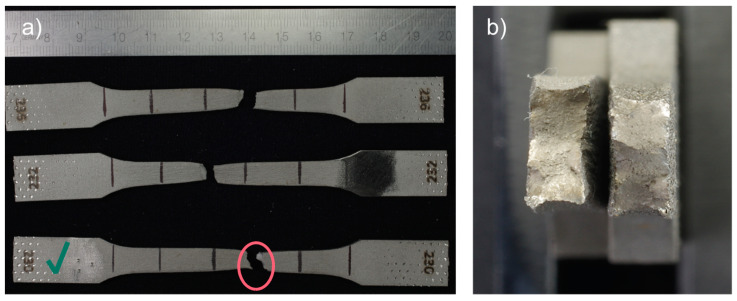
Fractures of tensile test EOS M 290 316L (HT) specimens presented as: (**a**) fractured specimens; (**b**) optical microscope image of the fracture surfaces, specimen 230. The samples have been numbered, which are visible in the figure. The letters L and R stand for the left-hand and right-hand sides of the sample in the figure. A green check mark means the specimen whose fracture faces were shown.

**Figure 10 materials-15-06278-f010:**
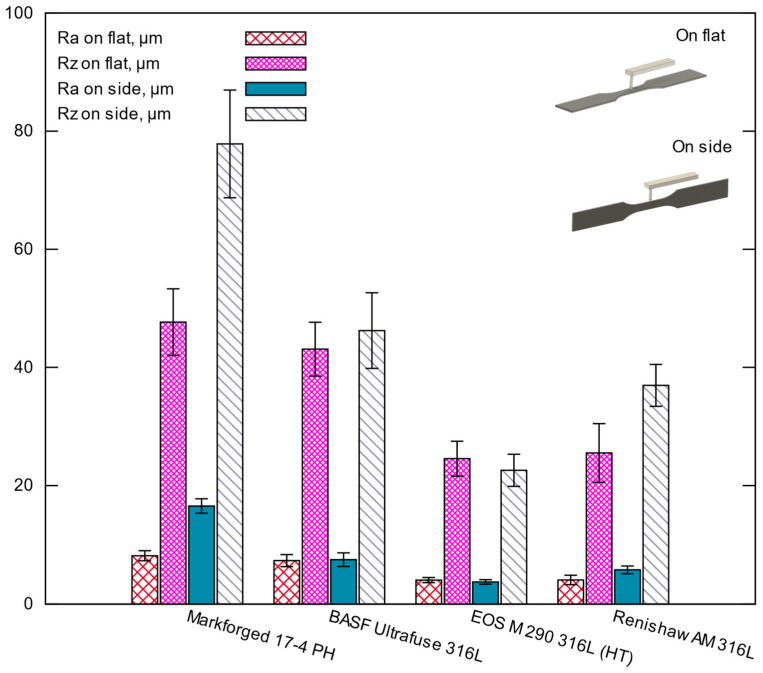
Results of roughness tests (mean ± standard deviation).

**Figure 11 materials-15-06278-f011:**
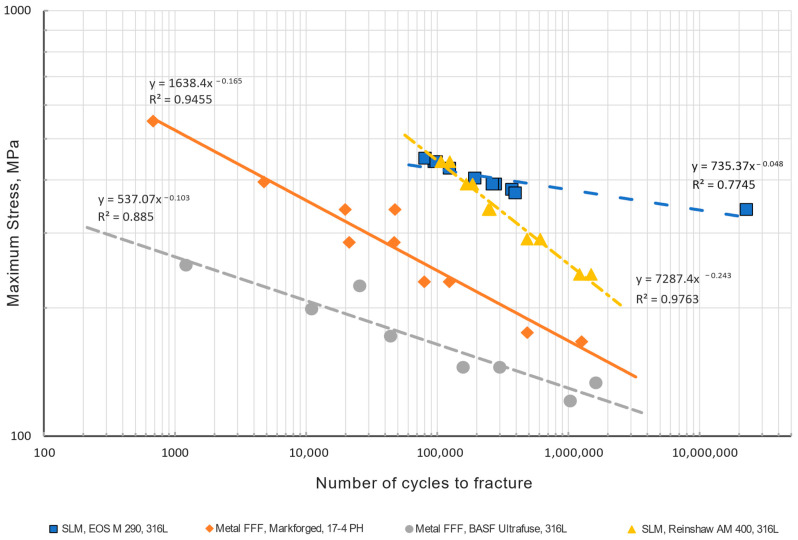
S-N curves for tested specimens; cycling tensile test with R = 0.1. (Two specimens of Ultrafuse 316L were broken in the grips, and their results were excluded.).

**Figure 12 materials-15-06278-f012:**
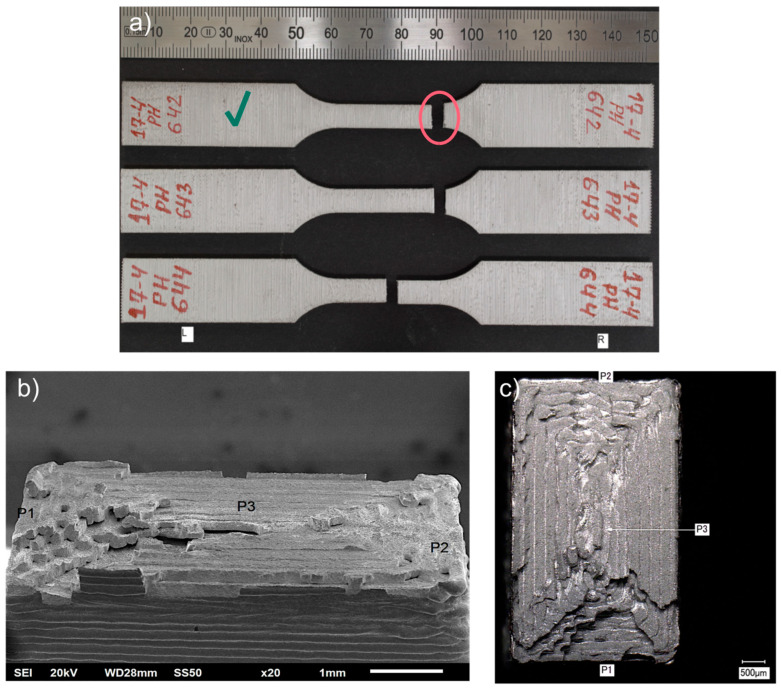
Fatigue fractures of Markforged 17-4 PH samples: presented as: (**a**) fractured specimens during the fatigue test; (**b**) fracture surface (SEM image), left side, specimen 642, at 4770 cycles; (**c**) top view of fracture surface (SEM image), left side, specimen 642, at 4770 cycles. The samples have been numbered, which are visible in the figure. The letters L and R stand for the left-hand and right-hand sides of the sample in the figure. A green check mark means the specimen whose fracture faces were shown.

**Figure 13 materials-15-06278-f013:**
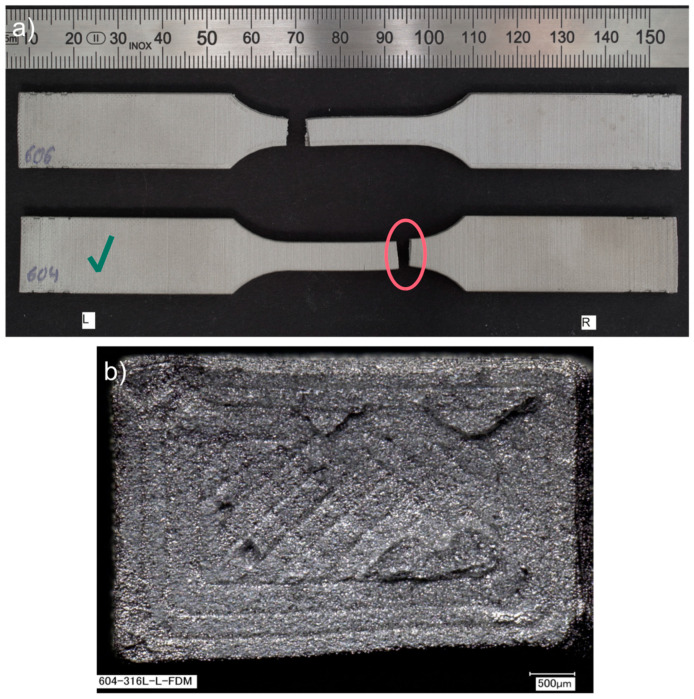
Fatigue fractures of BASF Ultrafuse 316L samples presented as: (**a**) fractured specimens during the fatigue test; (**b**) fracture surface (SEM image), left side, specimen 604, at 44,058 cycles. The samples have been numbered, which are visible in the figure. The letters L and R stand for the left-hand and right-hand sides of the sample in the figure. A green check mark means the specimen whose fracture faces were shown.

**Figure 14 materials-15-06278-f014:**
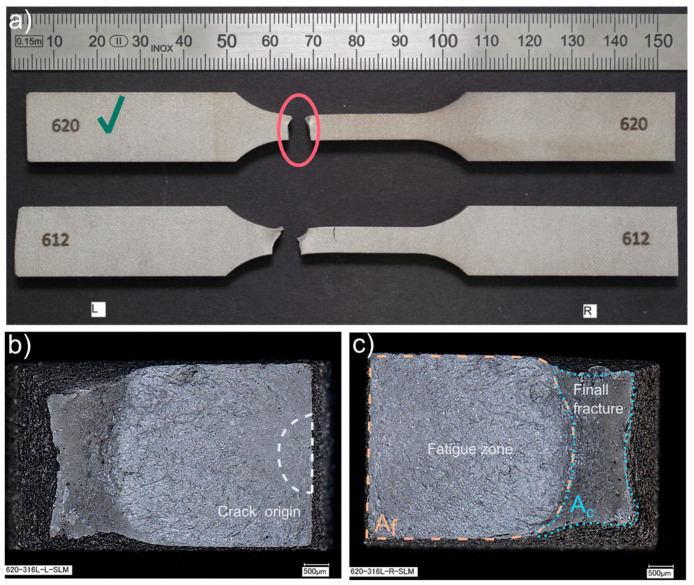
Fatigue fractures of Renishaw AM400 316L samples presented as: (**a**) fractured specimens during the fatigue tests; (**b**) fracture surface (SEM image), left side, specimen 620, at 1,492,546 cycles; (**c**) fracture surface (SEM image), right side, specimen 620, at 1,492,546 cycles. The samples have been numbered, which are visible in the figure. The letters L and R stand for the left-hand and right-hand sides of the sample in the figure. A green check mark means the specimen whose fracture faces were shown.

**Figure 15 materials-15-06278-f015:**
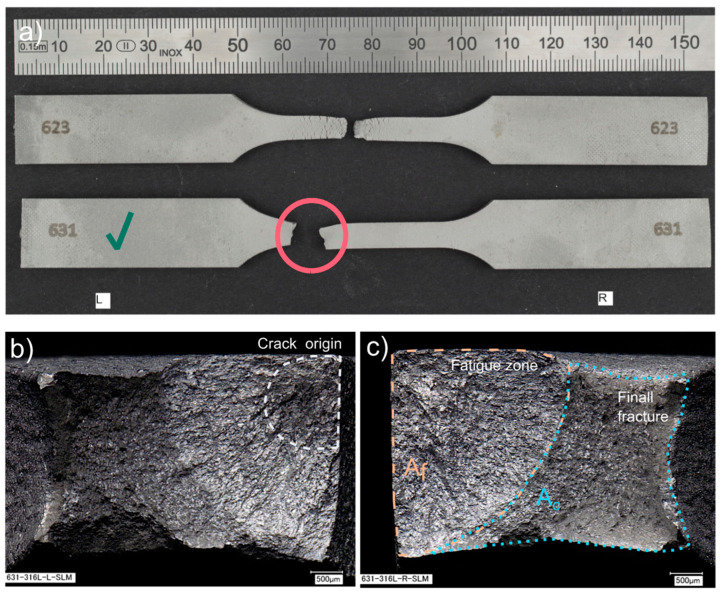
Fatigue fractures of Renishaw EOS M 290 316L samples: (**a**) fractured EOS M 290 316L specimens during the fatigue tests; (**b**) fracture surface (SEM image), left side, specimen 631, at 392,641 cycles; (**c**) fracture surface (SEM image), right side, specimen 631, at 392,641 cycles. The samples have been numbered, which are visible in the figure. The letters L and R stand for the left-hand and right-hand sides of the sample in the figure. A green check mark means the specimen whose fracture faces were shown.

**Figure 16 materials-15-06278-f016:**
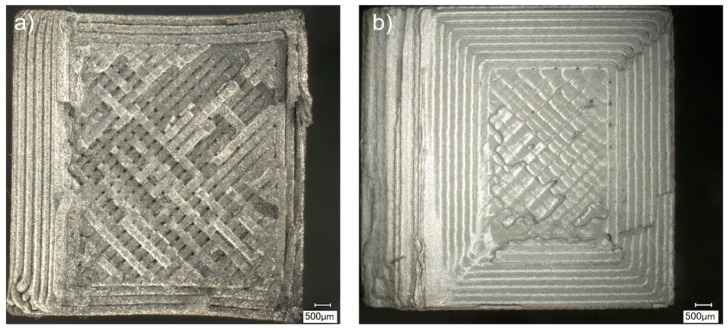
Fractured faces of the Charpy test metal FFF specimens presented as: (**a**) fractured surface (SEM image) of Ultrafuse BASF 316L specimen; (**b**) fractured surface (SEM image) of Markforged 17-4 PH specimen.

**Figure 17 materials-15-06278-f017:**
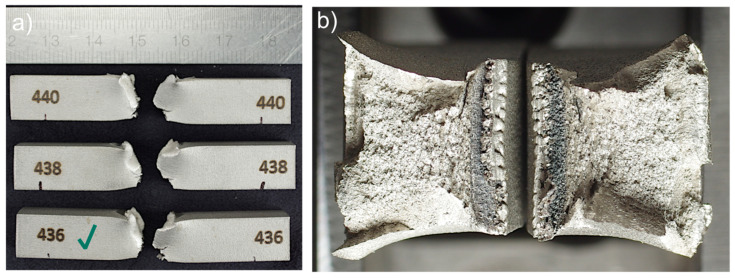
Fractures of Charpy test EOS M 290 specimens presented as: (**a**) fractured Charpy EOS M 290 specimens; (**b**) fracture surface (optical microscope image) of the EOS M 290 316L, specimen 436. The samples have been numbered, which are visible in the figure. The letters L and R stand for the left-hand and right-hand sides of the sample in the figure. A green check mark means the specimen whose fracture faces were shown.

**Table 1 materials-15-06278-t001:** Chemical composition of printed materials, (weight) wt %.

Elements	EOS 316L ^1^		Renishaw 316L [16] ^2^		Ultrafuse BASF 316L [17]		17-4 PH [18]	
	Min	Max	Min	Max	Min	Max	Min	Max
Fe	balance		balance		balance		balance	
C	-	0.03	-	0.03	-	0.07	-	0.07
Co	-	0.1	-	-	-	-	-	-
Cr	16	18	16	18	16	18.5	15	17.5
Cu	-	0.075	-	-	-	-	3.0	5.0
Mn	-	2.0	-	2.0	-	2.0	-	1.0
Mo	2.0	3.0	2.0	3.0	2.0	2.5	-	-
N	-	0.01	-	0.01	-	0.11	-	-
Nb	-	-	-	-	-	-	0.15	0.45
Ni	10	14	10	14	10	13	3.0	5.0
O	-	0.1	-	0.1	-	-	-	-
P	-	0.04	-	0.045	-	0.045	-	0.04
Si	-	1.0	-	1.0	-	1.0	-	1.0
S	-	0.03	-	0.03	-	0.03	-	0.03

^1^ Data form AnyShape (https://any-shape.com/) based on a made test. ^2^ Data according to EN1.4404 Stainless Steel (X2CrNiMo17-12-2).

**Table 2 materials-15-06278-t002:** Printing settings for the BASF Ultrafuse 316L specimens.

Printing Parameter	Value
Nozzle size, mm	0.4
Retraction distance, mm	0.5
Retraction speed, mm/s	45
Layer height, mm	0.15
Nozzle temperature, °C	250
Print bed temperature, °C	100
Chamber temperature, °C	100
Oversizing factors: X, Y, Z, %	21.35, 21.35, 26

**Table 3 materials-15-06278-t003:** Printing settings for the Markforged 17-4 PH specimens.

Printing Parameter	Value
Nozzle size, mm	0.45
Layer height, mm	0.125
Print bed temperature, °C	115
Metal hotend temperature, °C	220
Chamber temperature, °C	48
Oversizing factors: X, Y, Z, %	19.5, 19.5, 20

**Table 4 materials-15-06278-t004:** SLM process parameters used on EOS M 290 and Renishaw AM 400 printer.

Printing Parameter	EOS M 290	Renishaw AM 400
Layer height, µm	40	40
Laser power, W	*	180
Focus Diameter, µm	100	70
Hatch distance, mm	*	0.11
Shielding gas	Argon	Argon
Oxygen content, %	*	0.15

* Data not provided by AnyShape (www.any-shape.com) due to their confidentiality.

**Table 5 materials-15-06278-t005:** Results of tensile tests (mean ± standard deviation).

Parameters	Markforged 17-4 PH	BASF Ultrafuse 316L	EOS M 290 316L (HT)	Renishaw AM 400 316L
Yield strength, $R_{e}$, MPa	441.0 ± 55.4	152.9 ± 61.3	375.2 ± 22.4	443.1 ± 15.7
Tensile strength, $R_{m}$, MPa	495.9 ± 77.0	314.0 ± 41.7	533.2 ± 1.1	570.8 ± 4.4
Elongation at break A, %	0.4 ± 0.1	10.2 ± 5.0	38.2 ± 0.1	38.4 ± 0.1
Young’s modulus, E, GPa	142 ± 65	160 ± 22	190 ± 18	129 ± 17

**Table 6 materials-15-06278-t006:** Material properties as indicated by the manufacturer’s data sheets.

	Markforged 17-4 PH (XY) ^1^ [18]	BASF Ultrafuse 316L (XY/ZX) [17]	EOS M 290 316L ^2^ (XY/ZX) [23]	Renishaw AM 400 316L (XY/ZX) [16]
$R_{m}$, MPa	800	251/234	530/470	547/494
$R_{e}$, MPa	1050	561/521	640/540	676/624
A, %	5	53/36	40/54	43^+/−2^/35^+/−8^
E, GPa	140	-	-	197/190
ρ, density g/cm^3^	7.44	7.85	≥7.97	7.99
$a_{eN}$, impact strength, J/cm^2^	-	111/-	-	-
Hardness	30, HRC	128/128, HV10	-	198/208, HV0.5

^1^ No values for the Z direction are given by the manufacturer. ^2^ Values for as-built samples without heat treatment.

**Table 7 materials-15-06278-t007:** Average and standard deviation (mean ± standard deviation) of the surface roughness of fatigue specimens.

Roughness	Measurement Direction	Markforged 17-4 PH	BASF Ultrafuse 316L	EOS M 290 316L (HT)	Renishaw AM 400 316L
Ra,µm	Flat, along a sample	8.15 ± 0.85	7.34 ± 1.02	4.05 ± 0.43	4.07 ± 0.79
Rz,µm	Flat, along a sample	47.67 ± 5.64	43.09 ± 4.56	24.54 ± 2.95	25.51 ± 4.96
Ra,µm	Side, along a sample	16.55 ± 1.22	7.49 ± 1.15	3.73 ± 0.38	5.78 ± 0.65
Rz,µm	Side, along a sample	77.84 ± 9.12	46.23 ± 6.41	22.58 ±2.71	36.95 ± 3.55

**Table 8 materials-15-06278-t008:** Average and standard deviation (mean ± standard deviation) of the hardness of fatigue specimens close to fractures.

HB 2.5/187.5	Markforged 17-4 PH	BASF Ultrafuse 316L	EOS M 290 316L (HT)	Renishaw AM400 316L
	261 ± 17.4	126 ± 6.2	213 ± 9.7	217 ± 10.6

**Table 9 materials-15-06278-t009:** Average and standard deviation (mean ± standard deviation) of measured impact energy.

Markforged 17-4 PH, J	BASF Ultrafuse 316L, J	EOS M 290 316L (HT), J	Renishaw AM 400 316L, J
Not measurably small	Not measurably small	202.4 ± 17.3	223.3 ± 18.0

**Table 10 materials-15-06278-t010:** Overview of tensile properties for 316L stainless steel produced using metal FFF, SLM and MIM.

Method	Infill, Print Direction %	Build Orientation, Notes	Porosity ^7^ %	$R_{e}$ MPa	$R_{m}$ MPa	A %	E GPa	Source
FFF	100	XY	-	251	561	53	-	[17]
FFF	100	ZX	-	234	521	36	-	[17]
FFF	100 ^1^	XY	-	148	444	43	157	[27]
FFF	100	ZX	-	114	206	13	117	[27]
FFF	100	XY	1.5	167	465	31	152	[28]
FFF	100 ^2^	XY	8.2	-	421	43	-	[7]
FFF	100 ^2^	ZX	7.3	-	107	2.5	-	[7]
FFF	100 ^2^	XZ	8.5	-	356	28	-	[7]
MIM	N/A	-	5.3 ^6^	175 ^6^	520 ^6^	50 ^6^	-	[29]
MIM	N/A	water atomized	4	170	460	29	-	[30]
MIM	N/A	gas atomized	2	205	560	58	-	[30]
MIM	N/A	-	2 ^6^	180 ^6^	520 ^6^	40 ^6^	185 ^6^	[31]
SLM	100	XZ	2	412	577	35	139	[32]
SLM	100	ZX	2	365	469	17	78	[32]
SLM	100	XY	0.7	500	630	39	-	[33]
SLM	100	ZX	0.7	500	625	47	-	[33]
SLM	100 ^3^	ZX	2.3	512	622	20	-	[34]
SLM	100 ^3^	XY	2.3	430	509	12	-	[34]
SLM	100 ^4^	ZX	1.9	536	668	25	-	[34]
SLM	100 ^4^	XY	1.9	449	528	12	-	[34]
SLM	100 ^5^	XY	-	320	574	50	180	[35]
Wrought	N/A	hot rolled	-	241	621	59	185	[35]

^1^ Full density strategy [3]; ^2^ rectilinear infill pattern [15]; ^3^ single melt pattern [16]; ^4^ checkerboard melt pattern [16]; ^5^ print in the Y-direction; ^6^ typical values; ^7^ porosity is defined as (1—relative density). Where: $R_{e}—$yield strength, $R_{m}—$tensile strength, A—elongation at break, *E*—Young’s modulus.

**Table 11 materials-15-06278-t011:** Overview of tensile properties for 17-4 PH stainless steel produced using metal FFF, SLM and MIM.

Method	Infill, Print Direction %	Build Orientation, Notes	Porosity %	$R_{e}$ MPa	$R_{m}$ MPa	A %	E GPa	Source
FFF	100 ^1^	XY	9.8	443	497	0.79	108	[27]
FFF	100 ^1^	ZX	-	412	494	0.95	103	[27]
FFF	100 ^2^	XY	2.7	604	776	7.7	176	[38]
FFF	100 ^2^	XY	2.7	605	776	5.9	176	[38]
FFF	100 ^4^	XY aligned in Y	6.5 ^5^	580	794	2.7	128	[39]
FFF	100 ^4^	XY aligned in X	6.5 ^5^	600	795	3.2	131	[39]
FFF	100 ^4^	ZX	6.5 ^5^	647	701	0.76	134	[39]
FFF	100 ^4^	XY	-	688	1068	4.97	138	[9]
FFF	100 ^4^	XZ	-	650	815	0.86	189	[9]
FFF	100 ^4^	ZX	-	615	727	0.98	131	[9]
FFF	100	XY	1.4	746	1034	4.9	176	[40]
FFF	100	XZ	2.6	689	978	4.2	163	[40]
FFF	100	ZX	2.3	668	745	0.8	159	[40]
FFF	100 ^4^	XY	4	800	1050	5	140	[18]
FFF	100 ^4^	XY/H900	4	1100	1250	6	170	[18]
MIM	N/A	Heat-treated	4	965	1140	12	-	[41]
MIM	N/A	1038 °C @0.5 h	-	992	1018	13.4	199	[42]
MIM	N/A	H900	-	1387	1414	12.5	223	[42]
MIM	N/A	As sintered	3.2	730 ^3^	900 ^3^	6 ^3^	-	[29]
MIM	N/A	H900	3.2	965 ^3^	1070 ^3^	6 ^3^	-	[29]
SLM	N/A	ZX	-	830	887	61	133	[43]
SLM	N/A	ZX/H900	-	1050	1117	17	189	[43]
SLM	N/A	XZ	-	493	1058	19	-	[44]
SLM	N/A	XY/650 °C @1 h	-	428	1281	15	-	[44]
SLM	N/A	XY	-	535	1029	18	-	[44]
SLM	N/A	ZX	-	494	979	18	-	[44]
SLM	N/A	ZX/650 °C @1 h	-	483	1298	15	-	[44]
SLM	N/A	XY	-	635	1048	9.8	-	[36]
SLM	N/A	ZX	-	635	942	4	-	[36]
Wrought	N/A	-	-	980	1060	8	200	[45]
Wrought	N/A	-	-	1000	1103	5	-	[20]

^1^ Closed triangular cell path [3]; ^2^ printed in ZX and XY; ^3^ typical values; ^4^ metal X Markforged pathing arrangements [39]; ^5^ maximum porosity as a percentage of the area [39]. HXXX—means age-hardening treatment at XXX °F for 4 h air quenching.

**Table 12 materials-15-06278-t012:** Overview of high cycle fatigue properties for 316L and 17-4 PH stainless steels.

Alloy	Method	Infill %	Surface/Porosity %	Build Orientation, Notes	Max Stress MPa	Life Cycles	R	Source
316L	FFF	100	as sintered/4.4	XY	80 ^1^	***** 1.0 × 10^6^	0.1	[26]
316L	FFF	100	as sintered/4.4	XY	100 ^1^	1.04 × 10^5^	0.1	[26]
316L	FFF	100	as sintered/4.4	XY	120 ^1^	1.05 × 10^4^	0.1	[26]
316L	MIM	-	as produced/4	water atomized	90 ^1^	***** 3.32 × 10^6^	0.1	[30]
316L	MIM	-	as produced/2	gas atomized	135 ^1^	***** 1.0 × 10^7^	0.1	[30]
316L	SLM	-	Rz < 0.2 µm	H900	280 ^1^	3.0 × 10^6^	−1	[46]
316L	wrought	-	Rz < 0.2 µm	1100 °C, water	210 ^1^	***** 1.0 × 10^7^	−1	[46]
316L	wrought	-	Polished	-	438 ^2^	1.0 × 10^7^	−1	[47]
316L	wrought	-	Ra = 0.2 µm	1038 °C, air	220 ^1^	***** 1.0 × 10^6^	−1	[48]
316L	wrought	-	Ra = 0.2 µm	1038 °C, air	165 ^1^	***** 2.0 × 10^7^	0.1	[48]
17-4	MIM	-	as sintered/4	H1000	414 ^2^	***** 1.0 × 10^7^	−1	[49]
17-4	MIM	-	as sintered/2	H1000, HIP	448 ^2^	***** 1.0 × 10^7^	−1	[49]
17-4	SLM	100	Ra < 0.7 µm	ZX	225 ^1^	1.5 × 10^5^	−1	[50]
17-4	SLM	100	Ra < 0.7 µm	ZX/H900	280 ^1^	***** 1.0 × 10^6^	−1	[50]
17-4	SLM	100	Ra = 8.38 µm	ZX/CA-H1025	311 ^1^	***** 1.0 × 10^7^	−1	[51]
17-4	SLM	100	Ra = 0.015 µm	ZX/CA-H1025	480 ^1^	***** 1.0 × 10^7^	−1	[51]
17-4	SLM	100	Ra = 0.013 µm	ZX/CA-H1025	541 ^1^	1.266 × 10^6^	−1	[51]
17-4	SLM	100	Ra = 0.011 µm	ZX/HIP	560 ^1^	5.0 × 10^6^	−1	[51]
17-4	SLM	100	as produced	ZX	271 ^1^	***** 5.0 × 10^8^	−1	[52]
17-4	SLM	100	as produced	ZX/HIP	243 ^1^	***** 5.0 × 10^8^	−1	[52]
17-4	SLM	100	Machined	ZX	340 ^1^	***** 5.0 × 10^8^	−1	[52]
17-4	wrought	N/A	as produced	H900	355 ^1^	***** 5.0 × 10^8^	−1	[52]
17-4	wrought	N/A	Ra = 0.01 µm	H1025	750 ^1^	8.91 × 10^5^	−1	[51]

^1^ Tensile fatigue test. ^2^ Rotating beam fatigue tests, * runout fatigue test—a test specimen that just will not fail in the given amount of time. HXXX—means age-hardening treatment at XXX °F for 4 h air quenching.

## Data Availability

Data available on request due to restrictions eg privacy or ethical.

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
