# Peer review of "Strength Properties of 316L and 17-4 PH Stainless Steel Produced with Additive Manufacturing"

_materials, 2022, doi:10.3390/ma15186278_

Round 1

Reviewer 1 Report

The article highlights peculiarities of the properties of the material (316L stainless steel) printed using the Fused Filament Fabrication (FFF) method and in comparison with the material produced through Selective Laser Melting (SLM). The authors investigated the mechanical properties of these steels under static, dynamic and cyclic (fatigue) loading. However, the manuscript is written as a technical report (it does not contain a scientific analysis of the obtained results).

The following corrections must be made:

1.     Authors should pay attention to the display of link (Line 159).

2.     The section "Introduction" should be shortened, because the information provided in this section (Line 46-187) is too general and concerns purely technical characteristics of various methods of manufacturing parts in additive manufacturing.

3.     There are no references to Fig. 3 and Fig. 4.

4.     The authors should explain why the fatigue tests of the material obtained by the Fused Filament Fabrication (FFF) method were carried out, if the disadvantages of the FFF metal were obvious even during tensile tests.

5.     The authors should explain why the surface roughness of the studied samples was evaluated.

6.     The list of references does not meet the requirements of the journal, in particular, some references lack sufficient data (References 3, 4, 16, 19, 26, 43).

Author Response

Dear Reviewer,

Thank you for reviewing the article. I implemented the corrections which you pointed out as follows:

  1. The link issues are solved now.
  2. The introduction was shortened.
  3. The figure cross-references are corrected now.
  4. Concerning the fatigue tests of the material, I added a reason why I did them. (lines: 133-134, 140-147)
  5. Concerning roughness, I did explain why I made these measurements. (lines:479-482)
  6. The references were updated to meet the requirements of the journal.

Thank you

Slawomir Kedziora

Reviewer 2 Report

Many comparisons were performed in this manuscript, but no statistical tool was used. The authors must employ a statistical technique to compare your results to draw your conclusion.

Author Response

Dear Reviewer,

Thank you for your review of the article. Concerning your comments, I have added two figures, Fig. 5 and Fig. 10, showing the test result, where the mean values and standard deviations are shown. Now I believe it is more straightforward for readers to follow the text.

Thank you

Slawomir Kedziora

Reviewer 3 Report

Notes are included in the attached report

Author Response

Dear Reviewer,

Thank you for reviewing the article. I implemented the corrections which you pointed out as follows:

  1. The abstract and the article text are aligned in terms of objectives now.
  2. The introduction is shortened.
  3. All tables are created independently; all have cross-references in the text.
  4. I moved a part of the literature review to the discussion chapter because they are focused on test data and are closer to my test results. Additionally, the comparison between them is located there. I believe that it will now be more straightforward for readers.
    I did citations of 53 positions; most of them are from the last three years. Therefore, I believe I accurately presented the recent studies in this regard. I may have omitted something; if so, please indicate. Nevertheless, I updated the references a bit.
  5. All scientific symbols are explained now.
  6. I have added the comparison between the test and literature data in the Discussion section.
  7. The mathematical model was not planned with intention since the commercial solution of the metal FFF system (Markfoged Metal X) was used, which is "a close system". Control of the parameter is minimal. Thus, it is not possible to change parameters, which implies that it is not feasible to optimize the printing process.
  8. I am afraid I have to disagree with this comment; all samples that failed in the tensile machine grip were removed for testing. The shown specimens fractured in or at the boundary of the measuring zone (the gauge length).
  9. The conclusions were rewritten according to their importance.

Thank you

Slawomir Kedziora

Reviewer 4 Report

The paper seems interesting because it concerns the examination of the strength properties of stainless steel produced using rapid prototyping methods.  However, some minor corrections are required.

1.     The paper is too long. In particular, the introduction is too general and obvious. I propose to shorten it.

2.     In the text are some mistakes for example page 4, line 159 (see Error! Reference source not found.). Please correct it.

3.     Materials and methods section.

Please add information about grain size used in SLM technology. Furthermore in this section please add information about the roughness parameters used in the research.

4.     I propose to write the roughness parameters in the form Ra, Rz, not Ra, Rz.

5.     In the discussion section there lack of information about the interpretation of Rz roughness parameter. Please add it.

6.     In the paper I propose to add a table with the advantages and disadvantages of  FFF and SLM technologies.

Author Response

Dear Reviewer,

Thank you for reviewing the article. I implemented the corrections which you pointed out as follows:

  1. The introduction is shortened.
  2. The problem with "Error! Reference source not found." is solved now.
  3. After discussing the grain size data with all coauthors, we decided not to present it. Because we do not want to go for the details of microstructure, like defect size and shapes, defect distributions, and material structures. Those elements are essential to understanding material behaviour but can lead to different research topics in our case. We want to assess the usefulness of the metal FFF technology by showing strength properties, comparing them with the SLM technology, and displaying the root causes of differences. I hope that you will understand our point of view.
    The roughness parameters are now explained in the text. (lines: 309-315)
  1. The roughness parameters if a form of Ra, Rz is used in the whole text now.
  2. Rz parameter explanation and interpretation are now added to the text. (lines: 327, 489-491)
  3. Concerning the table, I agree that it can be added, but it can extend the article even more. Now, it is pretty long; therefore, I decided not to add it. I hope that you will understand my point of view. By the way, the text in the introduction contains this comparison.

Thank you

Slawomir Kedziora

Round 2

Reviewer 1 Report

The authors took into account all comments of the reviewer and made appropriate corrections to the manuscript.